*Report*

# RNase H1 and Sen1 ensure that transient TERRA R-loops promote the repair of short telomeres

Fabio Bento[1], Matteo Longaretti [1], Vanessa Borges Pires[2,3,4], Arianna Lockhart[2] & Brian Luke [1,2]✉

## Abstract

Telomere repeat-containing RNA (TERRA) is transcribed at telomeres and forms RNA–DNA hybrids. In budding yeast, the presence of RNA–DNA hybrids at short telomeres promotes homology-directed repair (HDR) and prevents accelerated replicative senescence. RNA–DNA hybrids at telomeres have also been demonstrated to prevent 5′end resection, an essential step for HDR. In accordance, we now demonstrate that, not only the presence, but also the removal, of RNA–DNA hybrids drives HDR at shortened telomeres during replicative senescence. Although RNase H2 is absent from short telomeres, it is quickly compensated for by the recruitment of RNase H1 and Sen1. The recruitment of RNase H1 is essential to allow for the loading of Rad51, consistent with the notion that RNA–DNA hybrids prevent Exo1-mediated end resection. In the absence of RNase H1 or Sen1 function, yeast cultures prematurely enter replicative senescence in the absence of telomerase. Furthermore, the delayed senescence phenotype observed when RNase H2 is deleted, depends on the presence of RNase H1 and Sen1. This study demonstrates the importance of transient RNA–DNA hybrids at short telomeres to regulate senescence.

**Keywords** RNA–DNA Hybrid; R-loop; RNase H1; Telomere; Senescence
**Subject Categories** Chromatin, Transcription & Genomics; DNA Replication, Recombination & Repair

## Introduction

Telomeres make up the ends of linear chromosomes and get transcribed into a long non-coding RNA referred to as TERRA (Azzalin et al, 2007). TERRA forms RNA–DNA hybrids that promote telomere repair and contributes to end protection, in yeast and human cells (Balk et al, 2013; Pfeiffer et al, 2013; Arora et al, 2014). In yeast, TERRA hybrids are regulated in a length dependent manner, whereby at long telomeres RNase H2 removes R-loops in S phase before the replication of the telomeres occurs (Balk et al, 2013; Graf et al, 2017; Moravec et al, 2016). When telomeres become critically short, RNase H2 association to telomeres is impaired, allowing hybrids to accumulate. The R-loops at short telomeres trigger homology directed repair (HDR) in a Rad51 and Rad52 dependent manner (Balk et al, 2013). R-loop-mediated telomere repair prevents the accumulation of critically short telomeres and, as a result, delays the onset of replicative senescence (Balk et al, 2013). Accordingly, the overexpression of RNase H1, which reduces R-loop levels, accelerates senescence, while the deletion of any subunit of the trimeric RNase H2 enzyme (Rnh201, Rnh202 or Rnh203) slows the rate of senescence (Graf et al, 2017).

It has recently been demonstrated, in yeast and human cells, that similar to telomeres, RNA–DNA hybrids and R-loops readily form at double-strand breaks (DSBs) and promote repair through the recruitment of HDR factors (D'Alessandro et al, 2018; Yasuhara et al, 2018; Teng et al, 2018; Marnef and Legube, 2021; Ohle et al, 2016). Importantly, at DSBs, the removal of the hybrids is just as important as their formation, highlighting the significance of timely R-loop regulation (Marnef and Legube, 2021; Ohle et al, 2016; Cohen et al, 2018). In human and yeast cells, the continued presence of RNA–DNA hybrids at DSBs prevents 5′end resection and, as a result, the accumulation of RPA and Rad51 on the resected strand is impaired (Ohle et al, 2016; Matsui et al, 2020; Yu et al, 2020). In accordance, following the formation of R-loops at DSBs, many proteins with R-loop resolving capabilities are subsequently recruited, including: SETX (Cohen et al, 2018), DDX5 (Yu et al, 2020) and RNase H1/2 (D'Alessandro et al, 2018), among others (Marnef and Legube, 2021). We have recently shown that the presence of persistent RNA–DNA hybrids at dysfunctional telomeres is able to reduce the rate, and extent, of Exo1-mediated resection of the 5′strand (Pires et al, 2023). Therefore, even though the accumulation of RNA–DNA hybrids promotes HDR at shortened telomeres in the absence of telomerase, we speculated that the hybrids must be subject to subsequent removal to allow efficient resection and HDR.

In this study, we demonstrate that there is a complex interplay between RNA–DNA hybrid resolvases at short telomeres. We confirm that RNase H2 binds to normal length telomeres but progressively loses association as telomeres shorten. RNase H1 (*RNH1*), on the other hand, is specifically recruited to telomeres when they become critically short, and this occurs as cells enter the S phase. The recruitment of RNase H1 to short telomeres promotes the loading of Rad51, likely by removing the RNA–DNA hybrid and facilitating Exo1-mediated resection. We believe that these results add an important level of regulation and understanding to how TERRA contributes to the repair

[1]Institute of Developmental Biology and Neurobiology (IDN), Johannes Gutenberg Universität, 55128 Mainz, Germany. [2]Institute of Molecular Biology (IMB), Ackermannweg 4, 55128 Mainz, Germany. [3]Instituto de Ciencias Biomedicas Abel Salazar, Universidade do Porto, 4050-313 Porto, Portugal. [4]Present address: Dortmund Life Science Center (DOLCE), Faculty of Chemistry and Chemical Biology, TU Dortmund University, Dortmund, Germany. ✉E-mail: brialuke@uni-mainz.de

of dysfunctional telomeres during replicative senescence. These data also have important consequences for cancer cells that employ the alternative lengthening of telomeres (ALT) mechanism, where TERRA RNA–DNA hybrids contribute to HDR and immortality (Arora et al, 2014; Azzalin, 2025).

## Results and discussion

### Transient R-loops prevent accelerated replicative senescence

The overexpression of *RNH1* can be used as a tool to remove persistent RNA–DNA hybrids. In the absence of telomerase, when yeast cells lose growth potential as telomeres shorten (Fig. 1A), *RNH1* overexpression accelerates the onset of replicative senescence in a manner epistatic with the loss of *RAD52*, a central recombination factor (Fig. 1B,C, compare blue and black lines) (Balk et al, 2013). These data confirm the importance of R-loops in promoting HDR at telomeres. Since RNA–DNA hybrids prevent Exo1-mediated end resection at dysfunctional telomeres (Pires et al, 2023), we speculated that the eventual removal of telomeric RNA–DNA hybrids must also be important for telomeric HDR. The overexpression of catalytically inactive *rnh1-D193N*, binds to, and stabilizes telomeric RNA–DNA hybrids (Wagner and Luke, 2022). Like *RNH1* overexpression, *rnh1-D193N* accelerated the rate of replicative senescence in telomerase negative cells (*est2*) (Fig. 1B compare orange and black lines), and did not have further effects

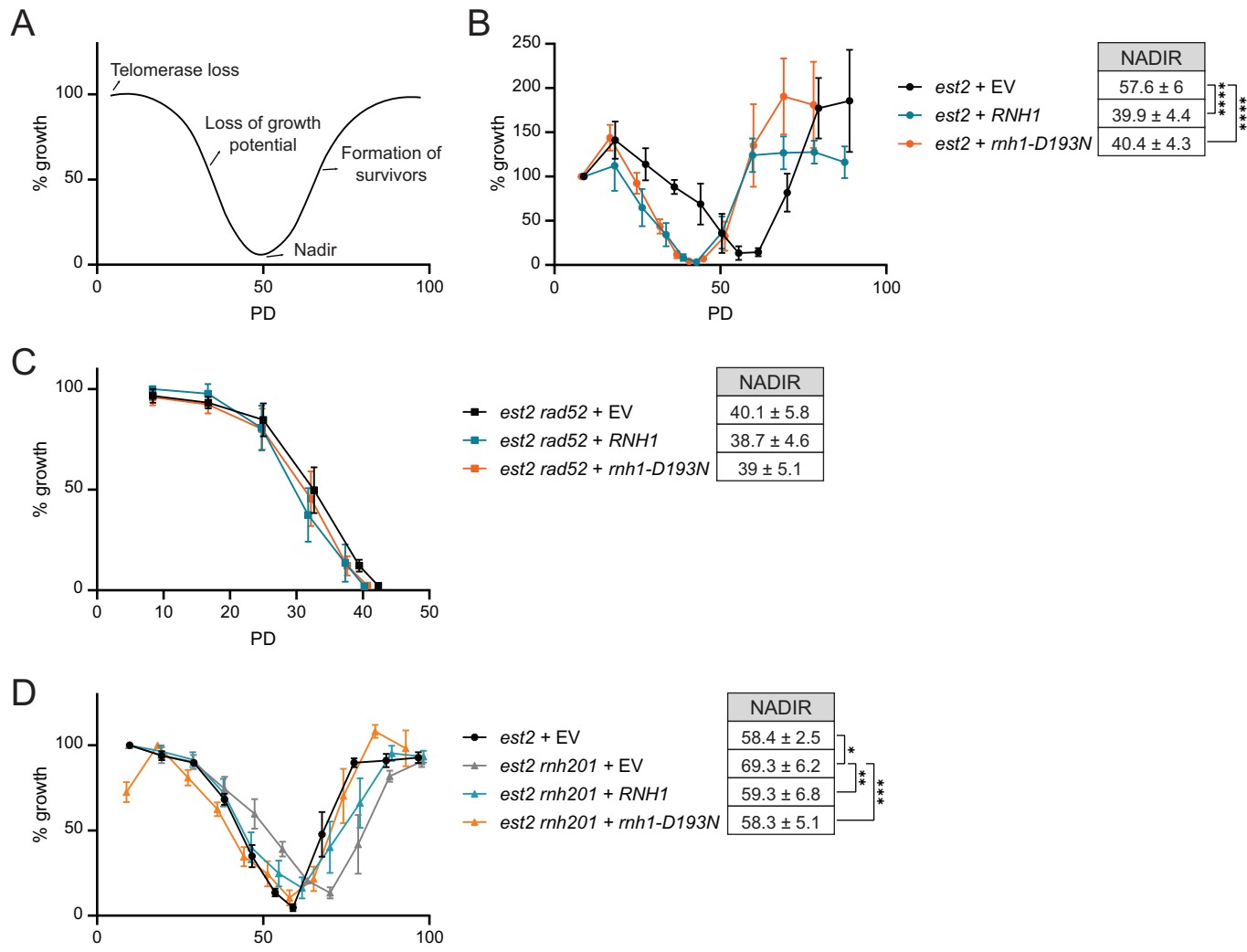

**Figure 1. RNA–DNA hybrids can promote, and prevent, replicative senescence.**

(A) Senescence curve scheme following telomerase loss. The nadir corresponds to population doubling (PD) at the average minimal growth potential before recombination-dependent survivors were formed. (B–D) Senescence curves were performed in telomerase deleted cells following the tetrad dissection of heterozygous diploids with the following genotypes (B) *est2*, (C) *est2 rad52*, and (D) *est2 rnh201* with the indicated plasmids. Following dilution and 24 h of growth (repeated daily), growth was estimated by measuring cell culture density with a spectrophotometer. The first or second measurement was set to 100%. Data shown as mean ± SEM; $n \geq 4$ biological replicates per genotype. PD = population doublings. Nadir data shown as mean PD ± SD; $n \geq 4$. *P*-values were obtained from unpaired t-test (*$p = 0.017$, **$p = 0.046$, ***$p = 0.022$, ****$p = 0.0003$). Source data are available online for this figure.

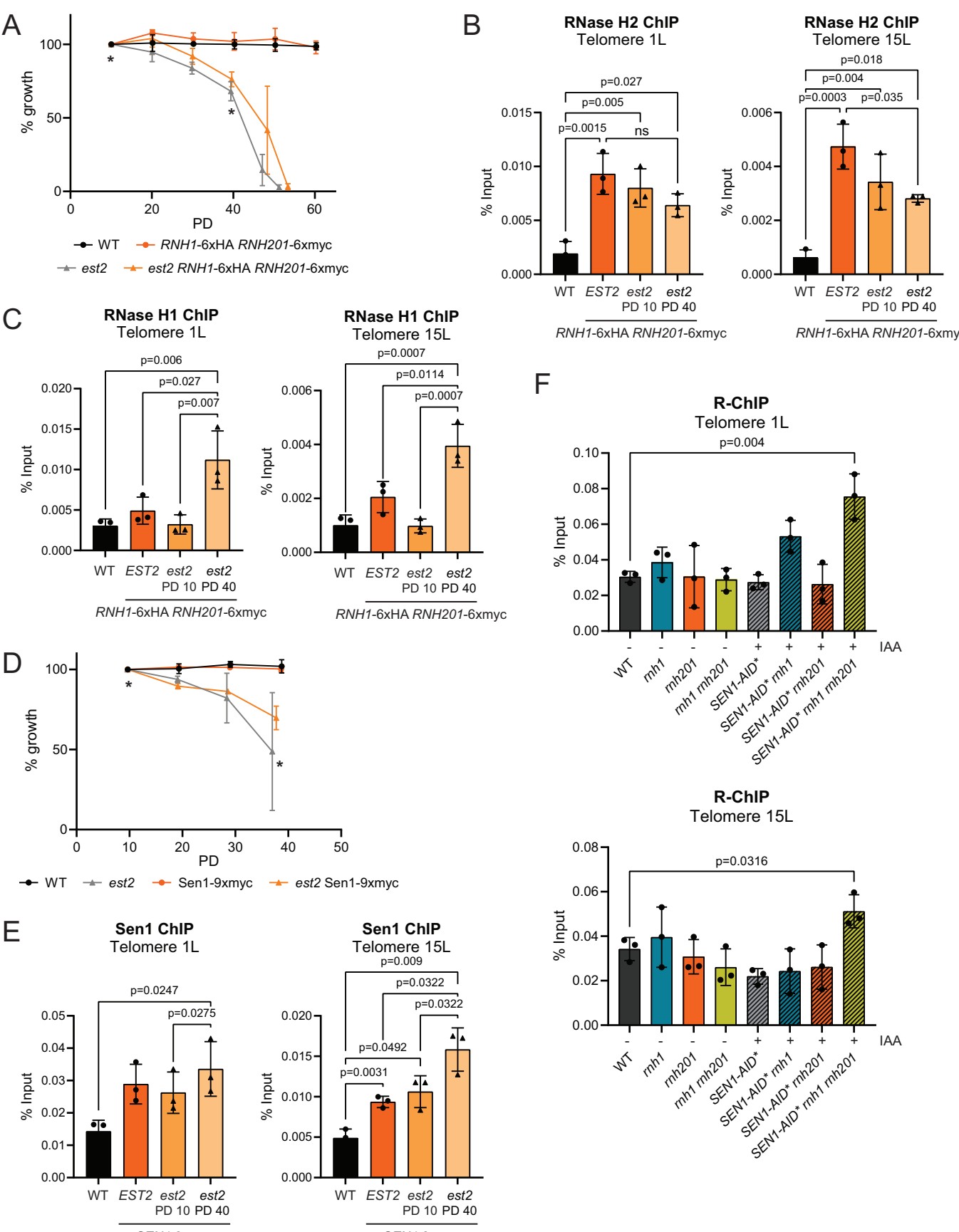

Figure 2. RNase H1 and Sen1 bind to telomeres in senescent cells.

(A) Senescence curves were performed in telomerase deleted cells (*est2*) with the indicated endogenously epitope tagged loci, and growth was estimated daily by measuring cell culture density, with the first measurement set to 100% as outlined in Fig. 1. Data shown as mean ± SEM; *n* = 3 biological replicates per genotype. The asterix (*) indicates when ChIP samples of exponential cells were collected; day 1 (PD 10) and day 4 (PD 40). PD = population doubling. (B, C) Rnh201 ChIP (B) and Rnh1 ChIP (C) were performed in exponential cultures of the indicated genotypes and population doubling (PD). Chromatin immunoprecipitation with HA or Myc antibody and qPCR analysis of the indicated strains at telomere 1 L and 15 L. Data shown as mean % input ± SD; *n* = 3 biological replicates per genotype. *P*-values were obtained from paired t-test. (D) Senescence curves were performed in telomerase deleted cells (*est2*) with the indicated endogenously epitope tagged loci, and growth was estimated daily by measuring cell culture density, with the first measurement set to 100% as outlined in Fig. 1. Data shown as mean ± SEM; *n* = 3 biological replicates per genotype. The asterix (*) indicates when ChIP samples of exponential cells were collected; day 1 (PD 10) and day 4 (PD 40). PD = population doubling. (E) Sen1 ChIP were performed in exponential cultures of the indicated genotypes and population doubling (PD). Chromatin immunoprecipitation with Myc antibody and qPCR analysis of the indicated strains at telomere 1 L and 15 L. Data shown as mean % input ± SD; *n* = 3 biological replicates per genotype. *P*-values were obtained from paired t-test. (F) R-ChIP performed in exponential cultures of the indicated mutants that overexpress *rnh1*(D193N)-3xHA transiently for 2 h. Sen1-AID* was degraded using 1 mM IAA for 5 h. Chromatin immunoprecipitation with HA antibody and qPCR analysis of the indicated strains at telomere 1 L and 15 L. Data shown as mean ± SD; *n* = 3 biological replicates per genotype. *P*-values were obtained from unpaired t-test. Source data are available online for this figure.

when *RAD52* was deleted (Fig. 1C). The effect of slowing the rate of replicative senescence through RNase H2 (*RNH201*) deletion is reversed through overexpression of either *RNH1* or *rnh1-D193N* (Fig. 1D). These results indicate that both the removal and the hyper-stabilization of RNA–DNA hybrids drive yeast cell populations into premature senescence. Therefore, although the loss of RNase H2 at short telomeres allows the accumulation of RNA–DNA hybrids to drive recombination (Graf et al, 2017), the eventual removal of these hybrids is of equal importance for HDR during replicative senescence. This mirrors what has previously been demonstrated at double-strand breaks (DSBs) whereby hybrids accumulate but then must be removed in order to allow 5′end resection (D'Alessandro et al, 2018; Ohle et al, 2016; Cohen et al, 2018).

## RNase H1 is only recruited to telomeres following telomerase loss

We speculated that either RNase H1 (Rnh1) or Sen1, or both, might be the RNA–DNA hybrid resolvase(s) that remove RNA–DNA hybrids following the loss of RNase H2 (Rnh201) at short telomeres. We epitope tagged all three RNA–DNA hybrid resolvases and allowed the cells to enter replicative senescence following the deletion of telomerase (Fig. 2A,D). The presence of the epitope tags had a minor effect on the rate of senescence compared to untagged *est2* cultures (Fig. 2A). Cross-linked samples were collected for chromatin immunoprecipitation (ChIP) at an early population doubling time (PD10) and a later time (PD40) to represent early, and late senescence, respectively. A wild-type strain without epitope tags served as a negative control for the ChIP (WT). As previously shown (Graf et al, 2017), Rnh201 is present at telomeres (1L and 15L) in telomerase positive cells (*EST2*), and it is progressively lost following the loss of telomerase with increasing population doublings (Fig. 2B). This effect is telomere-specific, as Rnh201 localization to the actin locus is not changed following telomerase loss (Fig. EV1A). Rnh1, on the other hand, was only found to localize to telomeres at PD40 (Fig. 2C), when Rnh201 levels were at their lowest levels at telomeres, and did not get recruited to the actin locus in late senescent cells (Fig. EV1B). This is consistent with the proposal that RNase H1 is only required at RNA–DNA hybrids when they become stabilized and lead to stress (presumably replication stress) (Lockhart et al, 2019; Zimmer and Koshland, 2016). To confirm that Rnh1 is specifically recruited to

short telomeres during DNA replication, we synchronized late senescent (PD40) cells in G1 with alpha-factor and released them synchronously into the cell cycle (Fig. EV1C,D) to perform ChIP before, during and after replication. Rnh1 association to telomeres increased as cells were released from G1 and entered S phase and decreased again when the bulk of replication had been completed (Fig. EV1E). The association of Rnh1 to telomeres coincided with the arrival of DNA polymerase epsilon (Pol2) to telomeres 30 min following the release from G1 (Fig. EV1F). Although the trends were consistent, there was variation between experiments as different senescent cultures had varying growth rates despite being matched for population doublings.

Similar experiments were performed with epitope tagged Sen1 and the same trend was observed (Fig. 2D,E). In contrast to Rnh1, Sen1 was already detectable at telomeres in telomerase positive cells (*EST2*) as compared to a non-tagged wild-type control (WT). Like Rnh1, Sen1 levels increased, albeit more subtly, at PD40 following telomerase deletion. This is consistent with the notion that Sen1 is a constituent component of the replisome (Appanah et al, 2020). The fact that Sen1 slightly accumulates in late senescence, may be indicative of replication stress/stalling, which has previously be demonstrated to occur more frequently at shortened telomeres in the absence of telomerase (Simon et al, 2016; Matmati et al, 2020; Dehé et al, 2012).

Together these results suggest that all three RNA–DNA resolvases work together to limit R-loops at telomeres. To confirm this, we performed R-ChIP at telomeres to measure RNA–DNA hybrids, as we have demonstrated that for yeast telomeres it is a more robust and reproducible method as compared to DNA-RNA immunoprecipitation (DRIP) with the S9.6 antibody (Wagner and Luke, 2022). In telomerase positive exponentially growing cells increased R-ChIP signals were only detected at telomeres when the function of all three RNA–DNA hybrid resolvases was eliminated (Fig. 2F). Taken together, RNase H1, RNase H2 and Sen1 combine to regulate RNA–DNA hybrids at telomeres with RNase H2 and Sen1 likely playing a predominant role. When telomerase is absent, RNase H2 is lost at telomeres and RNase H1 gets recruited to compensate.

## RNase H1 and Sen1 prevent accelerated senescence

Since the removal of hybrids during senescence is as important as their presence (Fig. 1), we speculated that Rnh1 and Sen1 would be

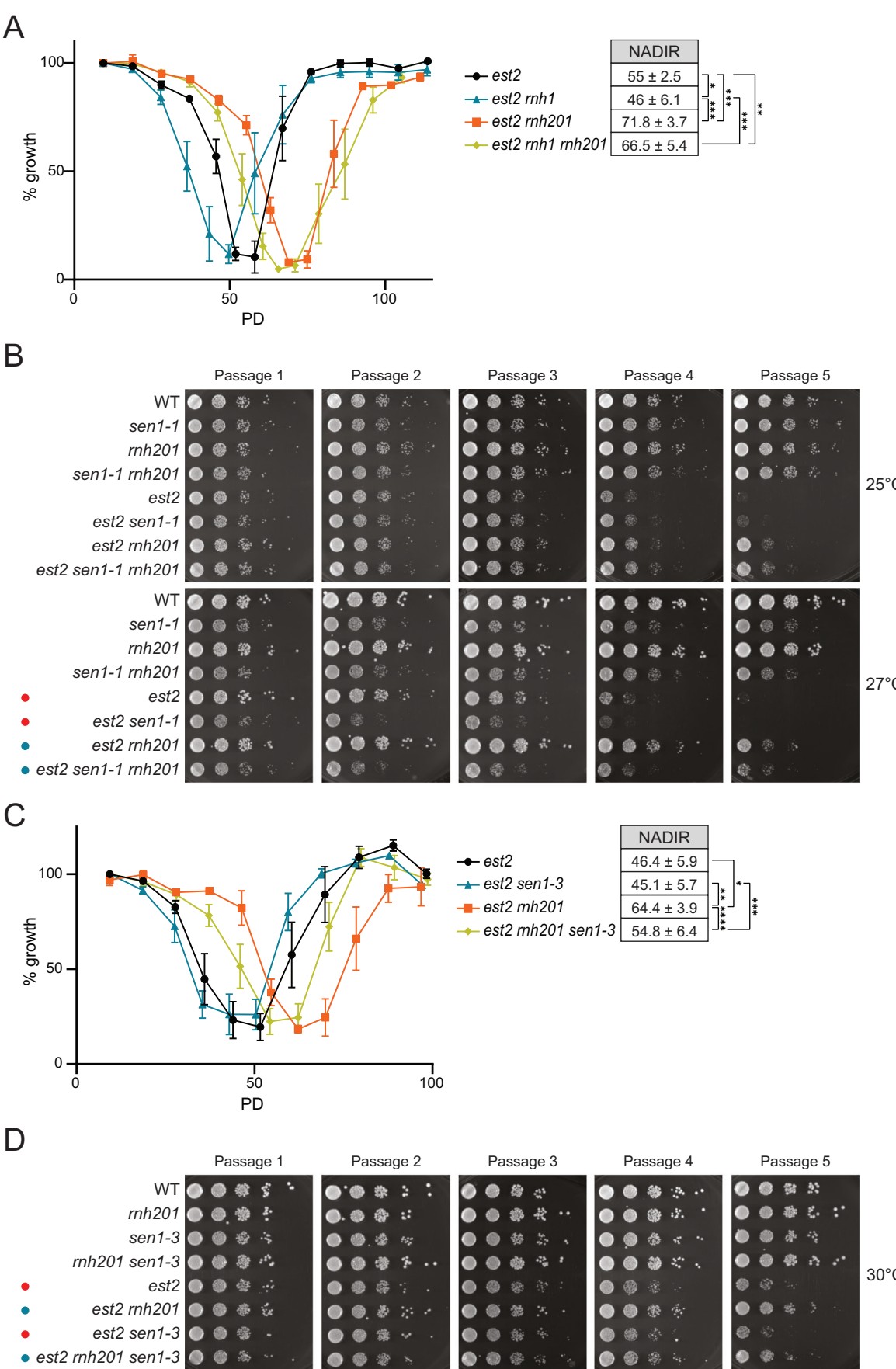

**Figure 3. RNase H1 and Sen1 prevent premature replicative senescence.**

(A) Senescence curves were performed in telomerase deleted cells (*est2*) with the indicated additional genotypes that were derived through tetrad dissection of heterozygous diploids. Viability was estimated daily by measuring cell culture density, with the first measurement set to 100%. Data shown as mean ± SEM; $n \geq 5$ biological replicates per genotype. Nadir data shown as mean PD ± SD; $n \geq 5$. P-values were obtained from unpaired t-test (*$p = 0.0313$, **$p = 0.0052$, ***$p < 0.0001$). (B) Senescence spotting—10-fold serial dilutions of the indicated strains were spotted onto YPD plates and incubated at the indicated temperatures. After 2 days, the first dilution was re-spotted onto new YPD plates for a new passage and placed at the indicated temperatures. Images were taken after 2 days growth. Images are representative of 5 biological replicates with 2 technical replicates or re-spotting. Red dots show the increased rate of senescence due to the *sen1-1* mutant, whereas blue dots highlight the reversal of slow senescence in the *rnh201* deletion. (C) Senescence curves were performed in telomerase deleted cells (*est2*) with the indicated additional genotypes that were derived through tetrad dissection of heterozygous diploids. Viability was estimated daily by measuring cell culture density, with the first measurement set to 100%. Data shown as mean ± SEM; $n \geq 5$ biological replicates per genotype. Nadir data shown as mean PD ± SD; $n \geq 5$. P-values were obtained from unpaired t-test (*$p = 0.0005$, **$p = 0.0001$, ***$p = 0.0253$, ****$p = 0.0218$). (D) Senescence spotting—10-fold serial dilutions of the indicated strains were spotted onto YPD plates and incubated at the indicated temperatures. After 2 days, the first dilution was re-spotted onto new YPD plates for a new passage and placed at the indicated temperatures. Images were taken after 2 days growth. Images are representative of 5 biological replicates with 2 technical replicates or re-spotting. Red dots show the increased rate of senescence due to the *sen1-1* mutant, whereas blue dots highlight the reversal of slow senescence in the *rnh201* deletion. Source data are available online for this figure.

required to prevent premature senescence. In accordance, the entry into replicative senescence was slightly accelerated when *RNH1* was deleted in addition to telomerase (Fig. 3A, black and blue lines). Furthermore, the delayed senescence that can be observed upon *RNH201* deletion, is partially reversed when *RNH1* is deleted in addition (Fig. 3A, orange and yellow lines).

To inactivate Sen1, we used the temperature sensitive *sen1-1* allele (Mischo et al, 2011) at permissive (25 °C) and semi-permissive (27 °C) temperatures. Since spontaneous suppressor mutations were frequent in the *sen1-1* liquid cultures, we performed the senescence assays on solid media with serial passaging. Similar to the loss of *RNH1*, we observed accelerated senescence in the *est2 sen1-1* double mutants compared to the *est2* single mutants at 27 °C (Fig. 3B, red dots). The delayed senescence upon *RNH201* loss was also clearly discernible on the solid media, and this was reversed in the *sen1-1* mutant background (Fig. 3B, blue dots). Although Sen1 functions in both transcription- and replication-associated processes (Appanah et al, 2020; Mischo et al, 2011; Aiello et al, 2022; San Martin-Alonso et al, 2021), we speculated that its replication role, in dealing with transcription-replication conflicts (TRCs), was critical for its effects during replicative senescence. We employed the *sen1-3* allele, which specifically loses interaction with the replisome, but maintains its transcription termination capabilities (Appanah et al, 2020). In the absence of telomerase, the *sen1-3* allele leads to a very slight accelerated senescence phenotype, but it reverses the delayed senescence in *est2 rnh201* cells (Fig. 3C,D, blue dots). In summary, both Rnh1 and replisome-associated functions of Sen1, play critical roles at shortened telomeres to deal with RNA–DNA hybrids.

## RNase H1 promotes Rad51 loading at short telomeres

The accumulation of RNA–DNA hybrids at telomeres impairs resection when the telomeres are rendered dysfunctional through Cdc13 inactivation (Pires et al, 2023). Following telomerase loss, critically short telomeres are subject to 5′end resection that is mediated by Mre11, Sae2, Sgs1, and Exo1, allowing for subsequent Rad51 and Rad52 loading onto the newly generated 3′ssDNA overhang (Fallet et al, 2014). Since Rnh1 is specifically recruited to short telomeres in late senescence, when RNase H2 is no longer present, we hypothesized that it may be required for R-loop removal, thereby allowing end resection and subsequent Rad51

loading. As expected, we could only detect Rad51 at telomeres in late (PD45), compared to early (PD10), telomerase negative senescent cells (Fig. 4A). Strikingly, Rad51 failed to accumulate efficiently at telomeres when *RNH1* was additionally deleted. At the actin locus (*ACT1*) there was no significant accumulation of Rad51, demonstrating that this was a telomere-specific regulation (Fig. EV2A,B).

Taken together, we propose a model that outlines how TERRA promotes HDR at short telomeres to prevent early onset replicative senescence (Fig. 4B). When telomeres are long (upper box), we have previously shown that TERRA is transcribed in a cell cycle-dependent manner (Graf et al, 2017). TERRA and R-loops are removed by Rat1 and RNase H2, respectively, before the replication machinery passes thereby avoiding replication stress. As Sen1 is constitutively associated with the replisome, it can resolve any TRCs that are not dealt with by RNase H2 in a timely manner. As telomeres shorten (Fig. 4B, bottom box), Rat1 and RNase H2 are lost at telomeres and RNA–DNA hybrids are transiently stabilized. We speculate that R-loop induced replication stress, and subsequent RPA associated ssDNA, may be the signal that promotes Rnh1 recruitment, given that Rnh1 is specifically recruited to short telomeres in the S phase (Figs. EV1E and 4B). This would also be in line with the interactions demonstrated between RNase H1 and RPA in human cells (Nguyen et al, 2017). Subsequently, Rnh1, together with Sen1, alleviate the TRC at the short telomere which would facilitate end resection and subsequent Rad51 loading. The short telomere is now competent for HDR and can be elongated from either the sister chromatid, or from another telomere.

Together, this study demonstrates that although the accumulation of RNA–DNA hybrids at telomeres is critical to drive HDR, the removal of the RNA–DNA hybrid is equally important. The question then arises as to why an RNA–DNA hybrid is required at all, if it simply needs to be resolved to get Rad51 loading. We speculate that the presence of an RNA–DNA hybrid, and hence replication stress, serves as a guide signal ensuring that only the critically short telomere is targeted for HDR. Once replication stress occurs, the short telomere can be processed for HDR, following the removal of the RNA–DNA hybrid. Given that TERRA can form RNA–DNA hybrids and replication stress in trans (Feretzaki et al, 2020), it will be interesting to determine if TERRA-based RNA therapy might be applicable to telomere biology disorders and promote telomere repair.

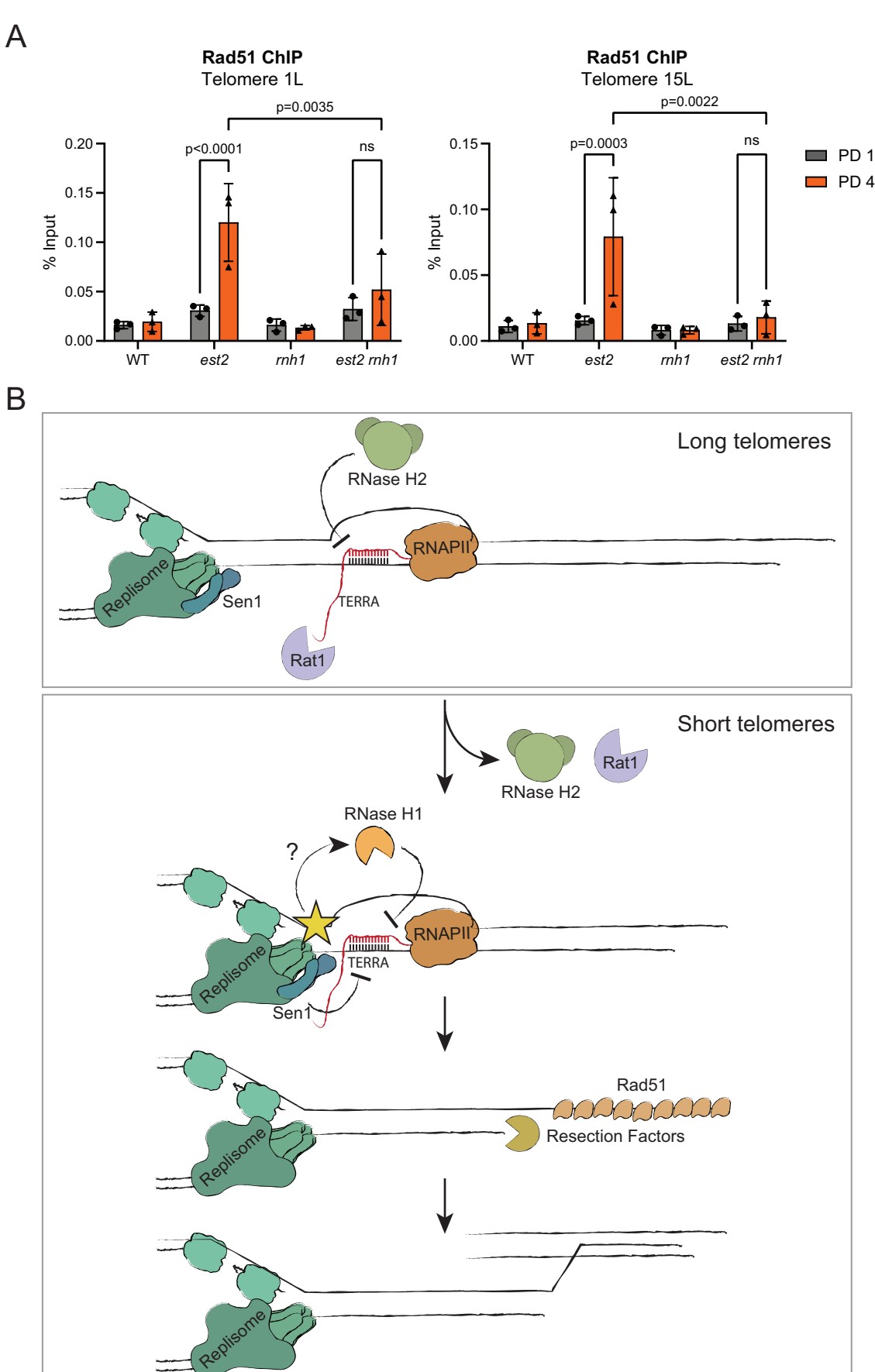

◀ **Figure 4. RNase H1 promotes efficient Rad51 loading.**

(A) Rad51 ChIP was performed in exponential cultures of the indicated mutants following 10 and 45 population doublings in the absence and presence of telomerase. Chromatin immunoprecipitation with Rad51 antibody and qPCR analysis of the indicated strains at telomere 1 L and 15 L. Data shown as mean ± SD; $n = 3$ biological replicates per genotype. $P$-values were obtained from two-way ANOVA ($p = 0.0035$, $p < 0.0001$, $p = 0.0003$, $p = 0.0022$). (B) At long telomeres, the TERRA transcript and resulting RNA–DNA hybrid that arise in early S phase are rapidly removed through the nucleases Rat1 and RNase H2, which likely prevents TRCs. These enzymes are lost at short telomeres and a TRC likely ensues. Although Sen1 is present at the replisome to resolve TRCs, Rnh1 is also recruited to help eliminate the RNA–DNA hybrid. Once the hybrid is removed, the 5'strand of the short telomere can be efficiently resected and Rad51 loaded. Together, this sequence of events promotes HDR at critically short telomeres and contributes to the avoidance of premature senescence. Source data are available online for this figure.

# Methods

## Reagents and tools table

| Reagent/Resource | Reference or Source | Identifier or Catalog Number |
|---|---|---|
| **Experimental models** | | |
| Yeast strains | This study | Table EV1 |
| **Recombinant DNA** | | |
| Plasmids | This study | Table EV2 |
| **Antibodies** | | |
| Anti-HA high-affinity (clone 3F10) | Roche | 11867423001 |
| Anti-Myc | Cell Signaling/NEB | 2276S |
| Anti-Rad51 | Abcam | Ab63798 |
| **Oligonucleotides and other sequence-based reagents** | | |
| PCR primers | This study | Table EV3 |
| **Chemicals, enzymes, and other reagents** | | |
| Gateway™ LR Clonase™ II Enzyme mix | ThermoFisher | 11791100 |
| Phusion High-Fidelity DNA Polymerase | ThermoFisher | F530L |
| Proteinase K | Genaxxon | M3037.0005 |
| RNase A | ThermoFisher | EN0531 |
| Agar | Sigma-Aldrich | 05040 |
| Bacto Yeast extract | BD Biosciences | 212750 |
| Peptone | AppliChem | A2210 |
| Alpha-factor | Zymo | Y1001 |
| Galactose | AppliChem | A3609 |
| Glucose | AppliChem | A1422 |
| Indole-3-acetic acid sodium salt (auxin) | Sigma-Aldrich | I3750 |
| Bradford solution | AppliChem | A6932,0500 |
| cOmplete, Mini, EDTA-free Protease Inhibitor Cocktail | Sigma-Aldrich | 4693159001 |
| Dynabeads™ Protein G | ThermoFisher | 10607605 |
| Formaldehyde 37% | Sigma-Aldrich | F8775 |
| MinELute PCR Purification kit | Qiagen | 28004 |
| **Software** | | |
| Graphpad Prism 8 | https://www.graphpad.com | |
| FlowJo | https://www.flowjo.com | |

## Experimental model and subject details

*Saccharomyces cerevisiae* strains used in this paper are derivatives of the standard strain S288C and are listed in Table EV1.

Strains were grown under standard conditions in YPD (Yeast Peptone Dextrose) or SC (Synthetic Complete) media at 30 °C if not indicated otherwise. Further specifications are mentioned within the Method Details and Figure legends section.

## Yeast strain generation

AID*-tagged strains were constructed as described by Morawska and Ulrich (Morawska and Ulrich, 2013). Yeast knock out strains were taken from yeast knock out collections (Winzeler et al, 1999). All plasmids used for strain construction are listed in Table EV2.

## Senescence curve

Telomerase negative spores of dissected diploids, grown for 3 days at the appropriate temperature after dissection, were inoculated to 0.01 $OD_{600}$ in 5 mL of appropriate media (YPD medium unless indicated otherwise) and incubated for 24 h at the appropriate temperature. The cultures' $OD_{600}$ was measured daily, and each culture was diluted to 0.01 $OD_{600}$ units in 5 mL of new media. Viability was measured by setting the starting culture $OD_{600}$ to 100% and comparing each daily measurement to the initial one, for each sample. 6 biological replicates were performed for each genotype, unless indicated otherwise. Population doublings (PD) were calculated daily as the $\log2(OD_{600}/0.01)$, where the $OD_{600}$ is the average cell density measured after 24 h for each genotype. PD values calculated do not account for the colony growth on the dissection plate (~25 generations). Graphs were plotted using Prism8 (GraphPad).

## Spotting assay

Overnight cultures of budding yeast were diluted in ten-fold serial dilution in sterile ddH₂O, starting from $OD_{600}$ 0.5. The yeast dilutions were spotted onto agar plates. The plates were incubated at 30 °C if not indicated differently. Images of the plates were taken after 24 h, 48 h, and 72 h using the Bio-Rad ChemiDoc™ Touch Imaging System (Colorimetric mode, 0.3 s exposure).

## Chromatin immunoprecipitation and R-ChIP

Chromatin immunoprecipitation (ChIP) experiments were in general conducted like described in our published method for

R-ChIP (Wagner and Luke, 2022). In brief, exponentially growing cells were cross-linked with 1.2% formaldehyde at room temperature for 10 min. Followed quenching with 115 mM glycine for 5 min and incubation on ice for at least 1 min. The samples were pelleted then at 4 °C and washed twice with 20 ml cold 1X PBS. After centrifuging, the pellets were stored at −80 °C.

The pellets were thawed on ice and resuspended in 400 µl cold FA lysis buffer -SOD (50 mM HEPES-KOH pH 7.5, 140 mM NaCl, 1 mM EDTA, 1% (v/v) Triton X-100, 1 protease inhibitor tablet per 10 ml buffer (Roche)). The samples were transferred then in lysing Matrix C tubes kept on ice and lysed with the FastPrep machine for 3 runs (each of 30 s) at 4 °C and level of 6.5 M/s, with 1 min on ice between runs. After that, the extracts were recovered by adding 800 µl cold FA lysis buffer plus SOD (50 mM HEPES-KOH pH 7.5, 140 mM NaCl, 1 mM EDTA, 1% (v/v) Triton X-100, 0.1% (w/v) Sodium deoxycholate, one protease inhibitor tablet per 10 ml buffer (Roche)). After mixing, the extracts were centrifuged for 15 min at 4 °C and the pellet was resuspended in 1.5 ml cold FA lysis buffer plus SOD. 20 µl 20% SDS were added and 750 µl of the resulting mix were combined with 0.4 g beads kept on ice. Sonication took place for 6 cycles of 30 s on/off at 4 °C with Bioruptor pico (Diagenode). The remaining volume of the mix was combined with 0.4 g beads kept on ice and sonicated in the same manner. The samples were centrifuged then for 15 min at 4 °C and the supernatant collected to a new tube, which constituted the ChIP extract.

Sonication efficiency was determined by evaluating the average length of the sheared DNA fragments. To accomplish this, 100 µl ChIP extract were combined with 100 µl elution buffer, de-cross-linked overnight at 65 °C, subjected to digestion with 7.5 µl Proteinase K and 1 µl RNase A and run on a 1.5% agarose gel for 45 min at 100 V.

To perform immunoprecipitation (IP), the protein concentration of the ChIP extract was measured by Bradford and diluted to 1 mg/ml in 1 ml cold FA lysis buffer plus SOD. From this mix, 50 µl were stored as 5% input at −20 °C. The proper amount of antibody (10 µl Anti-HA 3F10 (Roche) for R-ChIP and Rnh1 ChIP, 2 µl Anti-Myc (Cell Signaling) for Rnh2 and Pol2 ChIP, or 2 µl Anti-Rad51 (Abcam, ab63798) for Rad51 ChIP) was added to the IP samples, which were incubated for 30 min at 4 °C on a rotating wheel. 50 µl of the IgG Dynabeads, previously washed and supplemented with 5% BSA, were added and the samples were incubated overnight at 4 °C on a rotating wheel.

The IP samples were washed with 1 ml cold FA lysis buffer plus SOD, 1 ml cold FA lysis buffer 500 (50 mM HEPES-KOH pH 7.5, 500 mM NaCl, 1 mM EDTA, 1% (v/v) Triton X-100, 0.1% (w/v) Sodium deoxycholate), 1 ml cold buffer III (10 mM Tris-HCl pH 8.0, 1 mM EDTA, 250 mM LiCl, 1% (v/v) Tergitol-type NP-40, 1% (w/v) sodium deoxycholate) and 1 ml TE pH 8.0. After the last wash, the magnetic beads were eluted twice in 100 µl elution buffer, vortexed and incubated for 8 min at 65 °C. After passing the eluate to a fresh tube, 7.5 µl Proteinase K (20 mg/ml) were added to the IP samples, followed by overnight incubation at 65 °C. Meanwhile, the input DNA was thawed at room temperature, mixed with 150 µl elution buffer and 7.5 µl Proteinase K and incubated overnight at 65 °C. Following, both the immunoprecipitation and the input samples were purified with the Minelute PCR purification kit (Qiagen) and eluted in 50 µl water. The quantitative real-

time PCR was performed in a 10 µl total volume: 2 µl DNA, 5 µM of each primer, 5 µl of Eva green qPCR Master mix. The PCR was carried out under the following conditions: 10 min at 95 °C, 40 cycles of 15 s at 95 °C and 1 min at 60 °C, followed by a melting curve measurement (65.0 °C till 96.5 °C in 0.5 °C steps). All oligonucleotides used for quantitative real-time PCR are listed in Table EV3. The analysis of the data was carried out using the Bio-Rad CFX Manager software. The $C_q$ value of the input sample was corrected to account for the 1:20 dilution compared to the IP samples and the percent input of the IP samples was calculated:

$$\Delta C_q = C_q(corr.input) - C_q(sample \pm antibody)$$
$$Percent\ input = 100 * 2^{\Delta C_q}$$

### Cell cycle arrest and release

For arresting cells in G1, cells carrying mating type 'a' were treated with 2.4 µM alpha factor for 1.5 h to 2 h. The cells were either kept in G1 for further experiments or the alpha factor was washed out for release into the cell cycle. For the release, the cells were spun down for 3 min at 900 × g at RT and the supernatant was discarded. The cells were washed three times in one culture volume sterile ddH$_2$O and spun down again like before. The cells were resuspended in the same volume of YPD (pre-warmed to 25 °C) as the initial culture volume. The release was performed at 25 °C in a water bath shaker.

### DNA content flow cytometry

0.5 OD$_{600}$ units of exponentially cycling cells were spun down at 900 × g at room temperature (RT). The cells were washed in 1 ml ddH$_2$O and spun down as before. The cells were resuspended in 70% EtOH and stored at 4 °C overnight.

The samples were washed with 50 mM Tris-HCl (pH 7.4), resuspended in 500 µl 50 mM Tris-HCl with 0.25 mg/ml RNase A, and incubated at 37 °C for 3 h. 25 µl 20 mg/ml Proteinase K solution was added and the samples were incubated for 1 h at 50 °C. The samples were sonicated using a Branson Sonifier (10 s, constant mode, 10% power). 500 µl 50 mM Tris-HCl (pH 7.4) with 4 µM Sytox green added and the samples were measured using the BD FACS Fortessa flow cytometer. The analysis was carried out using the software 'FlowJo'.

## Data availability

This study includes no data deposited in external repositories.

The source data of this paper are collected in the following database record: biostudies:S-SCDT-10_1038-S44319-025-00469-7.

## Peer review information

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

## Acknowledgements

We thank the Luke lab members for support and discussions, the media lab, Flow cytometry, and protein production core facilities of the IMB. BL's lab is funded by the Deutsche Forschungsgemeinschaft (DFG, German Research Foundation) – Project-ID 491145305 - GRK 2859/1 and HA 6996/4-1. VP was supported by the Fundação para a Ciência e Tecnologia, Portugal (PD/BD/127999/20169).

## Author contributions

**Fabio Bento:** Conceptualization; Data curation; Formal analysis; Supervision; Funding acquisition; Investigation; Writing—original draft; Project administration; Writing—review and editing. **Matteo Longaretti:** Conceptualization; Data curation; Formal analysis; Investigation; Writing—review and editing. **Vanessa Borges Pires:** Formal analysis; Investigation. **Arianna Lockhart:** Investigation. **Brian Luke:** Conceptualization; Supervision; Funding acquisition; Investigation; Writing—original draft; Project administration; Writing—review and editing.

Source data underlying figure panels in this paper may have individual authorship assigned. Where available, figure panel/source data authorship is listed in the following database record: biostudies:S-SCDT-10_1038-S44319-025-00469-7.

## Funding

## Disclosure and competing interests statement

The authors declare no competing interests.

# Expanded View Figures

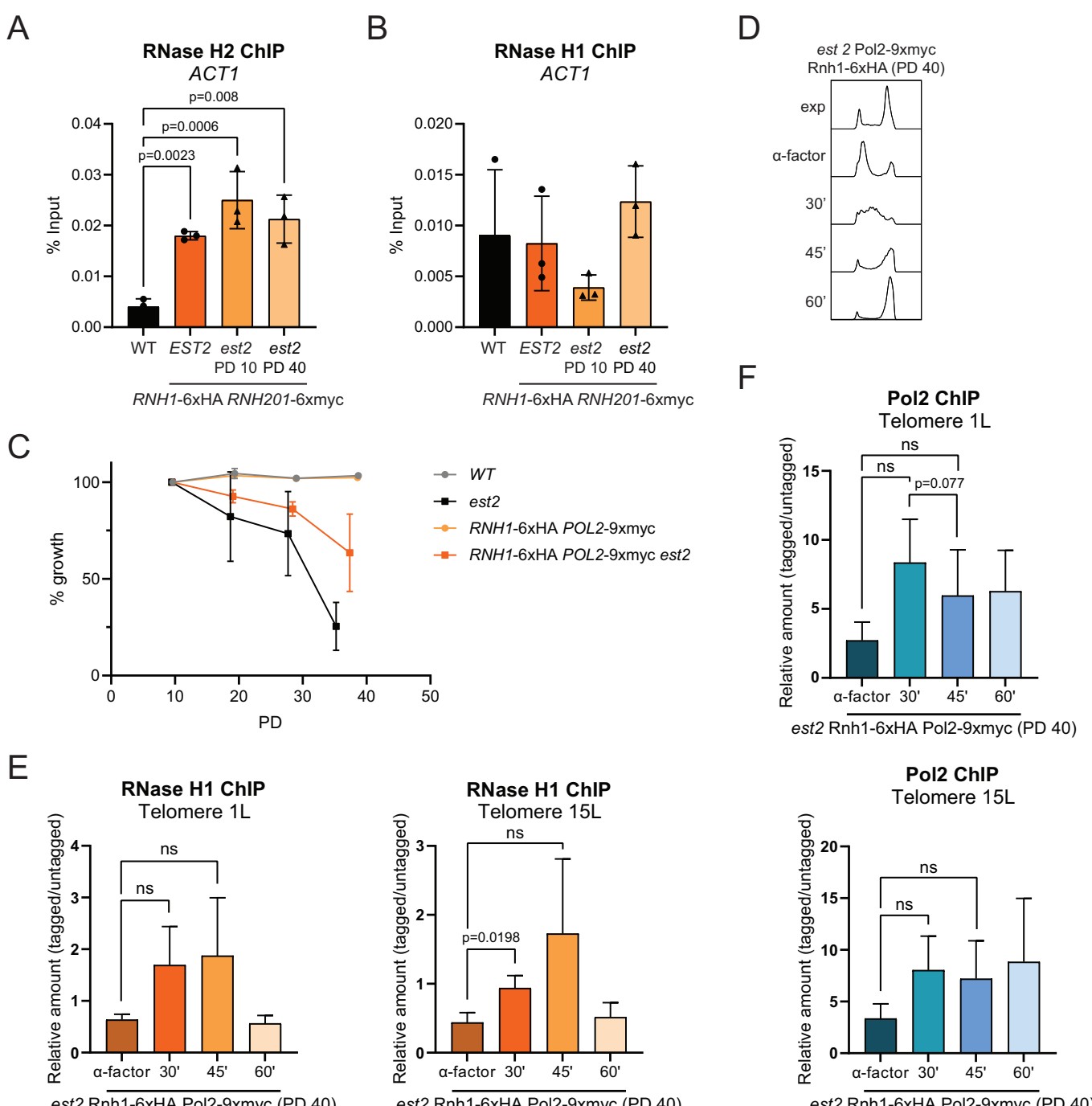

**Figure EV1. RNase H1 bind to telomeres in late S/G2 phase of senescent cells.**

(A) Rnh2 ChIP and (B) Rnh1 ChIP performed in exponential cultures of the indicated mutants. Chromatin immunoprecipitation with HA or Myc antibody and qPCR analysis of the indicated strains at Actin locus. Data shown as mean ± SD; *n* = 3. *P*-values were obtained from unpaired t-test. (C) Senescence curve was performed in telomerase defective cells (*est2*), and viability was estimated daily by measuring cell culture density, with the first measurement set to 100%. Data shown as mean ± SEM; *n* = 3 biological replicates per genotype. Samples were taken at day 1 (PD 10) and day 4 (PD 40) for DNA content analysis by flow cytometry (D) and ChIP (E, F). Cells were synchronized in G1 phase with α-factor for 2 h and released at 25 °C. (E) Rnh1 ChIP and (F) Pol2 ChIP performed at the indicated time points of the indicated mutants at telomeres 1 L and 15 L. Chromatin immunoprecipitation with HA or Myc antibody and qPCR analysis of the indicated strains at telomere 1 L and 15 L. Values are represented % input of DNA recovered and relative to untagged strains at correspondent time points. Data shown as mean ± SD; *n* = 3. *P*-values were obtained from two-way ANOVA.

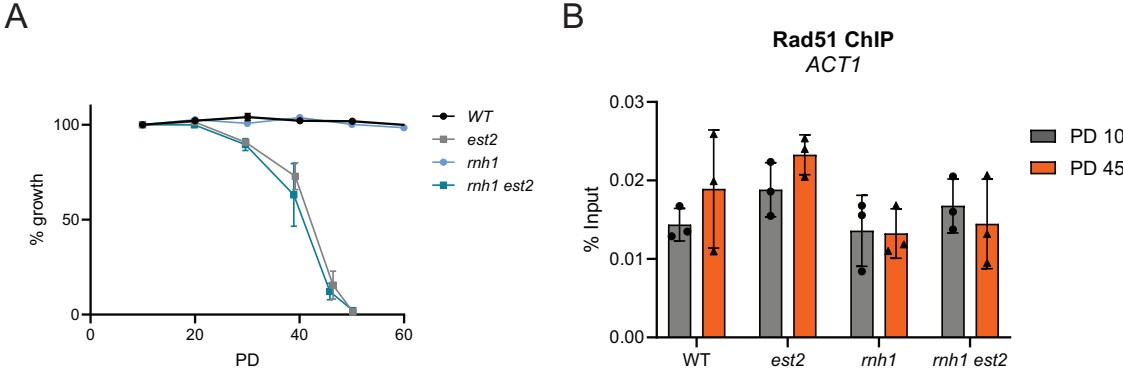

Figure EV2.  Rad51 does not bind to Actin locus.

(A) Senescence curve was performed in telomerase defective cells (est2), and viability was estimated daily by measuring cell culture density, with the first measurement set to 100%. Data shown as mean ± SEM; n = 3 biological replicates per genotype. Samples were taken at day 1 (PD 10) and day 5 (PD 45) ChIP (B). (B) Rad51 ChIP performed at the indicated time points of the indicated mutants. Chromatin immunoprecipitation with HA or Myc antibody and qPCR analysis of the indicated strains at telomere 1 L and 15 L. Values are represented % input of DNA recovered. Data shown as mean ± SD; n = 3. P-values were obtained from two-way ANOVA.

